



# Rare Earth Elements in oyster shells: provenance discrimination and potential vital effects

Vincent Mouchi[1], Camille Godbillot[1], Vianney Forest[2], Alexey Ulianov[3], Franck Lartaud[4], Marc de Rafélis[5], Laurent Emmanuel[1], Eric P. Verrecchia[6]

[1] Sorbonne Université, CNRS-INSU, Institut des Sciences de la Terre Paris, ISTeP, F-75005 Paris, France
[2] INRAP-Occitanie, UMR 5068, TRACES, Toulouse, France
[3] University of Lausanne, Institut des Sciences de la Terre, CH-1015, Lausanne, Switzerland
[4] Sorbonne Université, CNRS, Laboratoire d'Ecogéochimie des Environnements Benthiques, LECOB, F-66650, Banyuls, France
[5] Géosciences Environnement Toulouse, CNRS, IRD, Université Paul Sabatier Toulouse 3, 14 Avenue Edouard Belin, 31400 Toulouse, France
[6] University of Lausanne, Institut des Dynamiques de la Surface Terrestre, CH-1015, Lausanne, Switzerland

*Correspondence to*: Vincent Mouchi (vmouchi@gmail.com)

**Abstract.** Rare Earth Elements and yttrium (REY) in seawater originate from atmospheric fallout, continental weathering, and transport from rivers, as well as hydrothermal activity. Previous studies reported the use of REY measurements in biogenic carbonates as a means to reconstruct these surface processes in ancient times. As coastal seawater REY concentrations partially reflect those of nearby rivers, it may be possible to obtain a regional fingerprint of these concentrations from bivalve shells for provenance and environmental monitoring studies. Here, we present a dataset of 260 measurements of REY abundances by LA-ICP-MS from 42 oyster specimens from six locations in France (Atlantic Ocean and Mediterranean Sea), and from two species (*Crassostrea gigas* and *Ostrea edulis*). Our study reports that there is no significant difference in concentrations from shell parts corresponding to winter and summer periods for both species. Moreover, interspecific vital effects are reported from specimens from both species and from the same locality. REY profiles and t-distributed Stochastic Neighbour Embedding processing (t-SNE; a discriminant statistical method) indicate that REY measurements from *C. gigas* shells can be discriminated from one locality to another, but this is not the case for *O. edulis*, which presents very similar concentrations in all studied localities. Therefore, provenance studies using bivalve shells based on REY have to be first tested for the species, and are not adapted for *O. edulis*. Other methods have to be investigated to be able to find the provenance of some species such as *O. edulis*.

## 1 Introduction

Rare Earth Elements (REE) form a group gathering 15 elements (La to Lu) with similar electronic configuration of the atoms, similar properties and chemical behavior (Elderfield, 1988). The main sources of REE in seawater are the atmospheric fallout (Elderfield & Greaves, 1982; De Baar et al., 1983) and riverine input through continental weathering (Goldstein et al., 1984; Frost et al., 1986), as well as hydrothermal activity (Olivarez & Owen, 1991). In addition to these





various sources, the concentrations of REE in seawater are impacted by adsorption processes of REE to mineral surfaces and complexation (Sholkovitz et al., 1994; Schijf et al., 2015).

Reconstruction of REE compositions of seawater is generally used to provide information on past continental weathering, tectonic activity, and water mass circulation (Greaves et al., 1991; Censi et al., 2004; Haley et al., 2005; Piper & Bau, 2013). For example, REE profiles from continental shelf sediments are known to reflect those of the contributor rivers (Jouanneau et al., 1998). Moreover, specific elemental ratios from REE, such as Y/Ho, have been investigated as potential provenance proxies. Indeed, although the average Y/Ho value is equivalent to that of chondritic meteorites and mid-ocean-ridge basalts

(MORB; Jochum et al., 1986; Taylor & McLennan, 1988), it has been shown that Y/Ho fractionates in sediment particles, not only in seawater with depth, but also probably in waterbodies from watersheds (rivers and estuaries), whose composition has been modified depending on the weathered continental rocks (Bau et al., 1995; Nozaki et al., 1997; Prajith et al., 2015). The Y/Ho ratios in estuaries could therefore exhibit different values according to the regional inputs related to the mineralogical variability in continental covers.

It has been demonstrated that the seawater composition in REE and yttrium (REY) is recorded in carbonate materials, such as ooids (Li et al., 2019), brachiopod shells (Zaky et al., 2015, 2016), foraminifera tests (Osborne et al., 2017), and coral skeletons (Sholkovitz & Shen, 1995). In addition, the anthropogenic REE contamination of the Rhine River (Germany) has been demonstrated by the shell composition of freshwater mussels (Merschel & Bau, 2015). The reconstruction of the REY fingerprints in a coastal environment from mollusk shells could therefore be useful, not only to monitor potential

contaminations from anthropic activities (Le Goff et al., 2019), but also as a provenance proxy for quality control of cultured organisms prior to human consumption (Bennion et al., 2019; Morrison et al., 2019). This reconstruction can also be used in archaeology, since mollusk shells can be unearthed sometimes very far from the nearest shoreline (Bardot-Cambot, 2014), in order to rebuild historic trade routes. However, 'vital effects' (compositional shifts between inorganic and biogenic carbonate due to metabolic activity; Urey et al., 1951) have been reported to alter this regional fingerprint of REE in corals

(Akagi et al., 2004); therefore, a feasibility study needs to be conducted on mollusks before any extended archaeological perspective is considered.

To this end, REY measurements have been performed by laser-ablation inductively-coupled mass spectrometry (LA-ICP-MS) on modern and archaeological oyster shells from various French localities. The aim of this study is to assess the interspecific effects using specimens from two species: the flat oyster *Ostrea edulis* (Linnaeus, 1758), commonly found in

antique sites but still found on modern shores, and the cupped oyster *Crassostrea gigas* (Thunberg, 1793). Multiple measurements were performed on each of the specimens to evaluate the intraspecific variations, in particular regarding the potential seasonal fluctuations. A recent statistical method, the t-SNE, standing for "t-distributed Stochastic Neighbour Embedding" (van der Maaten & Hinton, 2008), is introduced as an attempt to discriminate the seven oyster groups of the study, in order to highlight significant interspecific and inter-regional differences between REY incorporations.



## 2. Material and methods

### 2.1 Modern-day settings and specimens

#### 2.1.1 Baie des Veys (Normandy)

The Géfosse area, in Baie des Veys (Normandy, France), is currently used as a commercial oyster farm location. This open sea area is characterized by a semidiurnal tidal range of 8 m from the British Channel and an overall siltation due to weaker ebb than flow, inducing a poor resuspension of sediment particles (Le Gall, 1970). The Baie des Veys is recharged in freshwater by the Isigny Channel, formed by the Vire and Aure rivers, and the Carentan Channel, constituted of the Douve and the Taute rivers, which drain a large part of the Bessin plain and the Cotentin. The watershed comprises limestone as well as basalt, acidic and alkaline metavolcanic rock and diorite (Baize et al., 1997). Respective flows of the Isigny and Carentan Channels are 19 $m^3$ $s^{-1}$ and 33 $m^3$ $s^{-1}$, these rates having no significant impact on the salinity of the shoreline (Sylvand, 1995).

Oyster specimens from this locality were gathered during a previous rearing experiment conducted between 2005 and 2006 (Lartaud et al., 2010a, 2010b; Mouchi et al., 2013). Although these specimens have been transplanted in several localities during their lives (for details, see Lartaud et al., 2010a), the part of the shells analyzed in the present study is restricted to that corresponding to their Baie des Veys stay. This period is recognized on the shells owing to *in vivo* chemical labeling performed during the rearing experiment (Huyghe et al., 2019). Both *Crassostrea gigas* (n=5) and *Ostrea edulis* (n=5) specimens from this locality and havig shared the same breeding location, have been considered (Figure 1, Table 1). *Crassostrea gigas* was recently renamed *Magallana gigas* by Salvi and Mariottini (2016); however, as this genus change is still debated (Bayne et al., 2017) and *C. gigas* being the most commonly found occurrence in the literature, this paper refers to this species by using its original genus name.

### 2.1.2 Leucate (Aude)

The Salses-Leucate lagoon is located on the southwestern French Mediterranean coast. It corresponds to a shallow coastal basin of 14 km long and 5 km wide, separated from the Mediterranean Sea by a sandy barrier interrupted by three narrow marine inlets. The average water depth is 1.7 m and the hydrology balances between entrance of marine waters from the Mediterranean Sea, supply of groundwater discharges from two main karstic springs with flows of 3 x $10^5$ $m^3$ $d^{-1}$ and 2 x $10^5$ $m^3$ $d^{-1}$, respectively (Fleury et al., 2007), and rainfall of 500 mm $d^{-1}$ restricted to the fall and spring periods. The superficial watershed covers 162 $km^2$ but the total area including the karstic waters is not yet known accurately, likely extended to 60 km far from the pool (Salvayre, 1989; Ladagnous & Le Bec, 1997), with karstic waters penetrating Jurassic and Cretaceaous limestone and dolomite. While tidal range is restricted, seawater level changes in the lagoon are controlled by strong northwesterly winds, regularly exceeding 10 m $s^{-1}$ (Rodellas et al., 2018).



*Crassostrea gigas* oysters (n=5; Figure 1, Table 1) originate from a wild brood stock in the vicinity of the local oyster farming area. We were not able to collect reliable *O. edulis* specimens from the Mediterranean Sea shoreline for comparison, as only aquaculture specimens of *C. gigas* are now available here.

### 2.1.3 Tès (Arcachon basin)

The Arcachon basin is a lagoon of 156 km$^2$ on the French Atlantic coastline. The area is subdivided into a subtidal zone and
an intertidal zone with a semidiurnal tidal range of 3 m, where the studied oysters grew. Freshwater is provided from a watershed of 4,138 km$^2$ by three main channels, the Eyre, the Porge and the Landes, as well as twenty-six other streams and local groundwater, for a total supply of 1,340 million m$^3$ freshwater per year (Lamour & Balades, 1979; Auby et al., 1994). The largest part of this watershed consists in Cenozoic river deposits, with, to a lower extent, limestone, clay (Dubreuilh & Bouchet, 1992), and some iron oxide deposits historically used as building material (Gourdon-Platel & Maurin, 2004).
*Crassostrea gigas* specimens (n=8; Figure 1, Table 1) from this locality originate from the same rearing experiment as those that were placed at Baie des Veys and measurements were restricted to the parts of the shells corresponding to the period spent at this locality (Lartaud et al., 2010a).

## 2.2 Archaeological sites and specimens

### 2.2.1 Lyon, Auvergne-Rhône-Alpes

In the area of the Fourvière hill of Lyon city, where remains of a building were inaccurately identified as a sanctuary to the goddess Cybele, a pit was filled by food wastes which included around 200 valves of the flat oyster *O. edulis* (Bardot-Cambot, 2013). Absolute dating of this pit is currently being re-evaluated, and is approximated to the beginning of the current era or during the 1st century CE. The provenance of these oysters is debated. Two groups of animals, one originating from the Mediterranean Sea coastline and the other from the Atlantic Ocean coastline, were identified based on
morphometric measurements and associated mollusc shells (Bardot-Cambot, 2013). Six *O. edulis* specimens were selected from the first group (later referred to as CYB-1 group) and seven more from the second group (CYB-2 group) for the preservation quality of their umbo (Figure 1, Table 1).

### 2.2.2 La Malène, Occitany

This medieval site (circa the 6th century CE) is located on top of a cliff and is constituted by the remains of a fortified
construction (Schneider & Clément, 2012). This *castrum* corresponds to one of the last antique sites where oysters were consumed by the elite, with supposedly spatially-restricted commercial travel (Bardot-Cambot & Forest, 2014). Still, the shells found in a dump, along with other proofs of the high social status of the occupants (such as golden currency and silver nails; Schneider & Clément, 2012), had been transported for over 120 km from the Mediterranean Sea. This origin is



certified because some valves are fixed on valves of *Flexopecten glaber*, which is endemic to the Mediterranean Sea. Six *O. edulis* specimens were selected for the preservation quality of their umbo (Figure 1, Table 1).

(Figure 1)

(Table 1)

## 2.3 Sample preparation

All specimens were mechanically cleaned from any epibiont and were selected according to the preservation state of their umbo region. The umbo was cut from the rest of the shell and embedded in Huntsman Araldite 2020 epoxy resin. Longitudinal thick sections (approx. 750 μm thick) were manufactured to expose the preserved internal structures (Figure 2), in order to perform geochemical analyses on this protected region, away from shell surface organic or chemical contaminants. An extensive chemical cleaning of the section surfaces, as advised by Zaky et al. (2015), was not performed as the analytical surface was preserved from any external contaminants over the history of the shell. However, the influence of organic matter occluded in the crystal lattice cannot be discarded, as any LA-ICP-MS work on biominerals.

(Figure 2)

All sections were observed under cathodoluminescence using a Cathodyne-OPEA cold cathode at ISTeP, Sorbonne Université (Paris, France). Observation settings were 15-20 kV and 200-400 μA mm$^{-2}$, at a pressure of 0.05 Torr. Areas potentially affected by diagenesis or damaged were identified in order to avoid any analysis of these regions by LA-ICP-MS. In addition, cathodoluminescence observations were used to define seasonal calibration of the umbo, according to Langlet et al. (2006) and Lartaud et al. (2010a). Only Leucate specimens did not have seasonal calibration as the CL signal was uniform and nearly absent for these specimens. A second seasonal calibration method from Kirby et al. (1998), based on a sclerochronological record on the ligamental area in the form of external convex and concave bands, was attempted. Unfortunately, umbos from Leucate specimens did not exhibit the necessary curved surface to conduct such a study. Consequently, measurement data from Leucate specimens were removed from the dataset used for the study of seasonal contrasts of the REY fingerprints.

## 2.4 Geochemical analyses

Chemical analyses were carried out by LA-ICP-MS at ISTE (University of Lausanne, Switzerland). Measurements were performed using an Element XR (ThermoScientific) ICP-MS coupled with a RESOlution 193 nm ArF excimer ablation system equipped with an S155 two-volume ablation cell (Australian Scientific Instruments). A pulse repetition rate of 20 Hz



and an on-sample energy density of 6 J cm$^{-2}$ were used. Pre-ablation of spots was first conducted in order to clean the surface of potential contaminants that could possibly be introduced during the sanding and polishing of the samples. The analytical spots were 200 µm in diameter. Ablation was performed on the areas of each sample section corresponding to winter and summer periods (according to the cathodoluminescence seasonal calibration). This protocol allows the REY incorporation to

be compared at different year periods throughout the life of the oysters (Figure 2c), with the exception of Leucate specimens, which did not exhibit seasonal cathodoluminescence signals. Sections from Leucate specimens were analysed at random positions over the umbo region instead. Multiple measurements were performed on each section to avoid bias from potential internal variability. Repeated measurements of NIST SRM 612 prior and following each 15-samples analytical series were used for external standardisation. Accuracy was checked against measurements of the BCR-2 basalt reference material from

USGS. Relative standard deviation from 22 measurements of BCR-2 was always better than 2.8% for all REY. Measured elements were La, Ce, Pr, Nd, Sm, Eu, Gd, Tb, Dy, Ho, Er, Tm, Yb, Lu, Hf, Y, and Ca as the internal standard. Data reduction was performed using the LAMTRACE software (Jackson, 2008). A total of 260 measurements were executed.

### 2.5 Data processing

Data processing was conducted using the Matlab software (MathWorks, www.mathworks.com, v. R2017a). None of the

measured elements exhibit a normal distribution (Kolmogorov-Smirnov test): the distribution is right-skewed, at larger element abundances. To facilitate further statistical treatment of such data, they need to be transformed to normality, for which several mathematical transforms can be used. Here, we use the cubic root transform (Chen & Deo, 2004). Seasonal differences (from the seasonal age models from cathodoluminescence) were estimated by hierarchical cluster analyses of 30 measurements from *C. gigas* specimens (n=5) and 41 measurements from *O. edulis* specimens (n=5) from Baie des Veys

(Appendix A). For both species, two methods for calculating cluster distances were tested, (i) unweighted average distance and (ii) "Ward" inner squared distance. A cophenetic correlation coefficient has been calculated, as it is the linear correlation coefficient between the distances obtained from the cluster tree and the original distances (in the multivariate space). It is an indicator of the accuracy of the distances (estimated on the tree) to faithfully represent the dissimilarities among the observations. Multivariate analysis of variance (MANOVA) was used to compare the REY fingerprints obtained from

measurements performed from *C. gigas* and *O. edulis* from Baie des Veys, against the null hypothesis that both datasets belong to the same population. Kruskal-Wallis tests were performed to compare the Y/Ho ratios of multiple groups, against the null hypothesis that all groups belong to the same population. A recent statistical method, the t-SNE (t-distributed Stochastic Neighbour Embedding; van der Maaten & Hinton, 2008), was used to compare and classify the multivariate dataset (exact Euclidean method). The idea is to embed high-dimensional data points in low dimensions in a way that

respects similarities between points. Nearby data points in the high-dimensional space correspond to nearby embedded low-dimensional points, and distant points in high-dimensional space correspond to distant embedded low-dimensional points (MathWorks, www.mathworks.com, v. R2017a).





## 3 Results

### 3.1 Comparisons of the inter- and intra-specific seasonal record

Firstly, heavy REE (Tm, Yb, Lu) and Hf were usually not detected (below 0.1 ng g$^{-1}$) in all the specimen groups, and were therefore removed from the dataset. Secondly, measurements from the Baie des Veys specimens allow for inter- and intra-specific comparisons, as they were performed on specimens from both *C. gigas* and *O. edulis* species. Data collected at the different seasons for each species did not show any significant difference in the incorporation of REY between winter and summer. Records from both winter (n=25) and summer (n=16) in *O. edulis* shell samples are mixed together, without

clustering samples with respect to seasons (Appendix A); in addition, the cophenetic correlation coefficients are 0.92 and 0.76 for average and Ward methods, respectively, which emphasizes the quality of the classification. Results are similar for *C. gigas* shells (n=19 and n=11 for winter and summer, respectively; Appendix A), with cophenetic correlation coefficients of 0.76 and 0.69 for average and Ward methods, respectively. However, a comparison between *C. gigas* and *O. edulis* (inter-specific comparison) clearly exhibits significant differences for both winter (MANOVA, p-value = 1.10$^{-8}$) and summer

(MANOVA, p-value = 2.10$^{-5}$) periods between the two species. Therefore, *C. gigas* and *O. edulis* seem to record differently their respective seasonal signals.

### 3.2 The Y/Ho ratio as a provenance proxy

The Y/Ho ratio is commonly used as a potential provenance proxy (Bau et al., 1995; Prajith et al., 2015). The obtained Y/Ho ratios during measurements on all the samples (Figure 3) do not display significant differences between the three localities

for *C. gigas* specimens (Kruskal-Wallis, p-values = 0.53 between specimens from Tès and Leucate, 0.87 between specimens from Tès and Baie des Veys, and 0.99 between specimens from Leucate and Baie des Veys). Also, all *O. edulis* modern and archaeological specimens from the other localities, except CYB-2, are similar to each other (Kruskal-Wallis, p-values = 0.96 between specimens from CYB-1 and La Malène, 0.97 between specimens from CYB-1 and Baie des Veys, and 1 between specimens from La Malène and Baie des Veys) but are different from *C. gigas* shells (Appendix B). Moreover, a significant

difference between *C. gigas* and *O. edulis* shells from Baie des Veys is also reported (Appendix B). The CYB-2 does not share the homogeneity of other *O. edulis* populations. Indeed, the ratios measured in *O. edulis* specimens from this group are not significantly different from the ones obtained in modern *C. gigas* specimens (Kruskal-Wallis, p-values = 0.93, 0.19, and 0.39, when compared with specimens from Tès, Leucate, and Baie des Veys, respectively).

(Figure 3)

### 3.3 REE incorporation and dispersion in shells

For all the specimens, a gradual decrease in REE dispersion is generally observed with their increasing atomic number (Figure 4). Indeed, several groups can be identified with light REE (LREE; *e.g.*, Pr and Sm), such as the *C. gigas* groups. On

the contrary, medium REE (*e.g.*, Dy and Er) distributions appear similar for all groups, with only the Leucate specimens being clearly separated from the other locations. The REE median profiles (normalized to the Post-Archean Australian Shales; McLennan, 1989) also present similar trends of medium REE (from Gd to Er) for most groups (Figure 5), except for Leucate specimens, which exhibit lower abundances than the other groups for all REE. However, light REE are generally substantially depleted in *C. gigas* specimens compared to *O. edulis* groups (approx. one order of magnitude difference). The

only exceptions are La and Ce in *C. gigas* shells from Baie des Veys, which present values in the range of those from *O. edulis* specimens. This particularity of enriched Ce (and to a lesser extent, La) is not shared by *O. edulis* specimens from this same locality. Although these two elements have similar abundances for both species in this locality, all the other REE abundances are different.

(Figure 4)

(Figure 5)

Results from the entire dataset (*i.e.*, 260 measurements and 12 elements per measurement) are compared in Figure 6 using t-

SNE. Some groups are well identified by this method, such as the three groups of *C. gigas* from Tès (Atlantic coastline), Leucate (Mediterranean Sea coastline) and Baie des Veys (British Channel coastline). The *O. edulis* groups are however not discriminated by t-SNE. In this sample set, it appears that CYB-1 and Baie des Veys specimens are relatively similar in terms of range of distribution on one hand, and that CYB-2 and La Malène specimens share similarities on the other hand; but the four groups remain poorly differentiated.


(Figure 6)

### 4 Discussion

In both studied species, the decrease in the range of variation of REE abundances with the increasing atomic number (except

for Tm, Yb, Lu, and Hf, which were not quantified in the shells) can be explained by the increased affinity of heavy REE (HREE) for complexation in seawater, as it has been demonstrated in previous studies (Cantrell & Byrne, 1987; Byrne & Kim, 1990; De Baar et al., 1991). As these elements are trapped into complexed forms or ligands, their bioavailability in seawater is strongly reduced, limiting their insertion in the oyster ionic pumps leading to the mineralisation locus. These





bioavailability restrictions of REE have already been demonstrated in the freshwater mussel *Corbicula fluminea* for Gd
(Merschel & Bau, 2015) and other REE (Ponnurangam et al., 2016). Another explanation can be advanced regarding the
technique used. LA-ICP-MS device analyses both the mineral and organic phases ablated from the biomineral without the
possibility to assess their relative proportions. Although the mean proportion of organic compounds in oyster shells is limited
(<0.5% for *C. gigas*; Mouchi et al., 2016), it is known that organic REE abundances are depleted in HREE (Freslon et al.,
2014). The decreasing abundance with the increasing atomic number may then be caused by protein and polysaccharide
contents. Only extensive cleaning for solution-based ICP-MS analyses would be able to remove entirely the organic
molecules before measurements, but this would not fit the fast REE assessment from a large number of specimens we aimed
to conduct in this study.

In this study, the Y/Ho ratios, which are usually proposed as a provenance proxy (Bau et al., 1995; Prajith et al., 2015), are
affected by strong vital effects for both oyster species, potentially due to the decrease in REE abundance with increasing
atomic number. In addition, significant differences between species from the same locality are also reported. Hence, the
Y/Ho ratio generally does not depend on the original location (Figure 3). Consequently, Y/Ho should not be used directly as
a provenance proxy (at least from LA-ICP-MS data collected on biogenic carbonates), or with extreme caution after having
discarded any potential vital effect. As the decreased REE abundance with increasing atomic number discussed above
prevents a locality-specific variation of Ho, other Y/REE ratios have been tested as alternative provenance proxies using
lighter REE (Figure 4). Y/La, Y/Ce, Y/Pr and Y/Nd were all unsuccessful to provide identification of locality groups and
also present similar values for all *O. edulis*. For Y/La and Y/Ce ratios, *C. gigas* specimens from Baie des Veys were identical
to all *O. edulis* specimens, while for Y/Pr and Y/Nd ratios, *C. gigas* specimens from Baie des Veys were identical to *C. gigas*
specimens from Tès. Overall, Y/REE ratios appear unsuccessful for provenance discrimination.

Measurements performed on Baie des Veys' specimens have been used to study the incorporation of REE inside a single
species, in order to evaluate its intraspecific variation. Contrary to the models performed for different temperatures on the
mussel *Mytilus edulis* (Ponnurangam et al., 2016), the seasonal conditions do not have any impact on the REE incorporation,
neither for *O. edulis* nor *C. gigas* shells (in the range of 5-20 °C; Table 1). Other parameters than temperature and pH, used
by Ponnurangam et al. (2016), are probably in effect, which lowers the impact of temperature on REE incorporation. This
observation implies that any part of a shell can be sampled without necessarily having to define a temporal calibration of the
umbo. However, REE abundances fluctuate widely within a single specimen, and the "local fingerprint" of these elements
needs to be clarified and based on multiple measurements performed on each of several specimens. These intra-individual
fluctuations cannot be due to seasonally-controlled environmental factors, such as temperature, precipitations or plankton
blooms. However, a potential source of REE for oysters can be the porewater or resuspended sediment (Haley et al., 2004;
Crocket et al., 2018), and therefore, the relative abundances may fluctuate without precise temporal cyclicity.

The reasons for these intra- and inter-specific vital effects remain unknown. Indeed, as far as we know, no study has ever
shown evidence or suspicion of the use of REE in metabolic processes that could induce an effective filter of these elements
between seawater and the extrapallial cavity where shell mineralization occurs. Nevertheless, it has been reported that REE,



or other unsuspectedly useful elements, are indeed used by organisms in specific environmental settings; an example is provided by diatoms in Zn-depleted conditions, where Zn is used as a co-factor of carbonic anhydrase (Lee et al., 1995).

Another example is given by methanotrophic archaea, which use Cd, a toxic element, as a co-factor of methanol dehydrogenase (Pol et al., 2014).

Incorporation of REE differ between both studied species. Not only *C. gigas* shells present different REE profiles for each group (unlike those of *O. edulis*), but also the Gd positive anomaly, a characteristic of modern coasts under pressure of anthropic activities (Bau & Dulski, 1996; Nozaki et al., 2000; Le Goff et al., 2019), is observed solely for "modern" *C. gigas*

specimens from Baie des Veys and Tès (Figure 5). The Gd anomaly is neither visible in modern *O. edulis* from Baie des Veys nor in modern *C. gigas* from Leucate. In the latter, this may be due to the fact that (i) most of the freshwater input in the Leucate area originates from karsts and (ii) the watershed does not include any major city; consequently, these distinct settings are less inclined to anthropogenic Gd conveying, which is mainly related to modern rivers crossing towns where magnetic resonance imaging is in use in hospitals (Le Goff et al., 2019). The systematically low abundances of REE in the

Leucate shells can also be explained by the regional geology, as watersheds of the other localities of *C. gigas* specimens present substratum types with higher REE contents than karsts (*e.g.*, such as basalts). Overall, these species-specific characteristics indicate that *C. gigas* can be used as a sentinel species regarding REE pollution of coastal waters. On the contrary, *O. edulis* is not a proper candidate for such studies.

Several reasons for these interspecific differences can be advanced. It is known that oysters can be selective in their diet,

composed mainly of diatoms (Yonge, 1928; Paulmier, 1971) of a specific size range, and preferentially digest specific species of diatoms over others (Shumway et al., 1985; Cognie et al., 2001). If food is a source of REE, it may be possible that each oyster species does not feed on the same prey, which can present different abundances of these elements. Aquarium experiments reported different ingestion rates in the 5-15 µm algal size range between these oyster species (Nielsen et al., 2017), but there is no indication on the REE content of the food. Alternatively, *O. edulis*, which exhibits generally higher

abundances of REE (nearly one order of magnitude higher for light REE, except for La and Ce for Baie des Veys *C. gigas* specimens; Figure 5), could present a higher bioaccumulation of these elements in its soft tissues (and eventually shell) compared to *C. gigas*. Ong et al. (2013) presented trace element measurements from soft tissues of both species from the Baie de Quiberon (Brittany, France) indicating that soft tissues of *O. edulis* contain generally less Cu and Zn but more Cd and Pb than those of *C. gigas*. Such species-specific bioaccumulation and incorporation differences could also be in effect

for REE. Finally, it is possible to explain the higher abundance of LREE in *O. edulis* shells compared to *C. gigas* by suggesting that *O. edulis* ingests more clay particles. As HREE are trapped in complexed form in seawater, mainly LREE must be available for adsorption on clay particles, and eventually integrate the forming carbonate shell.

In any case, this study shows that t-SNE can be used on REY measurements from *C. gigas* shells to identify regions of origin of specimens from this species. However, it appears that intraspecific vital effects prevent its efficiency on other oyster

species, such as *O. edulis*, which specimens exhibit the same fingerprint for several localities of origin. For this reason, we cannot confirm or refute a different origin of the two populations of the Cybèle archaeological specimens (CYB-1 and CYB-

2). Oysters are not an exception. Indeed, such similar interspecific vital effects had previously been reported as well for corals (Akagi et al., 2004).

## 5 Conclusions

Multiple types of vital effects on REE incorporation in *C. gigas* and *O. edulis* oyster shells have been highlighted in this study. Intraspecific variations in REE abundances are significant but not related to seasonal fluctuations. A gradual decrease in REE incorporations with increasing atomic numbers has been observed, and it appears that HREE are less discriminant than LREE to identify the various studied groups. The Y/Ho ratio, previously reported as a proxy for provenance studies, remains ineffective in oyster shells. Finally, interspecific variations underline the ability of t-SNE procedure to correctly

separate *C. gigas* specimens of various origins but not *O. edulis* specimens, which implies that only *C. gigas* can be used as a monitor species of LREE pollution. Reconstruction of provenance of oyster specimens will therefore have to be performed separately for each studied species, as regional geochemical fingerprints of the shells appear to be species-dependant. In order to be able to identify the regions of origin of archaeological remains corresponding to species affected by strong vital effects (such as *O. edulis*), it is necessary to investigate other chemical elements as potential provenance proxies.

**6 Data availability**

The data used in this manuscript have been deposited on the Zenodo data repository (Mouchi et al., 2019).

## 7 Author contribution

VM conceived the study. VM, CG and AU performed the data analysis. VF and FL provided specimens. VM and EV performed statistics and data processing. VM, CG, MdR, LE, EV and FL interpreted the results. VM wrote the manuscript

with contributions from all authors.

## 8 Competing interests

The authors declare that they have no conflict of interest.

## 9 Acknowledgements

The authors would like to thank Frédéric Delbès for the work he performed on the preparation of the thin sections.




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





| Group | Species | Locality of origin | Coordinates | Age | Annual temperature range | Annual salinity range (PSU) | Number of specimens | Number of measurements |
|---|---|---|---|---|---|---|---|---|
| TES | *C. gigas* | Tès, Arcachon (Atlantic Ocean) | 44°40.01 N, 01°08.18 W | Modern | 5-26°C | 25-35 | 8 | 58 |
| CYB-1 | *O. edulis* | Unknown | Unknown | 20-30 CE | Unknown | Unknown | 6 | 44 |
| CYB-2 | *O. edulis* | Unknown | Unknown | 20-30 CE | Unknown | Unknown | 7 | 30 |
| LMP | *O. edulis* | Unknown | Unknown | 6th c. CE | Unknown | Unknown | 6 | 43 |
| Leucate | *C. gigas* | Leucate, Aude (Mediterranean pond) | 42°52.48 N, 03°01.50 W | Modern | 2-32°C | 26-42 | 5 | 14 |
| BDV-gigas | *C. gigas* | Géfosse, Baie des Veys (Normandy) | 49°23.11 N, 01°06.05 W | Modern | 5-20°C | 30-34 | 5 | 30 |
| BDV-edulis | *O. edulis* | Géfosse, Baie des Veys (Normandy) | 49°23.11 N, 01°06.05 W | Modern | 5-20°C | 30-34 | 5 | 41 |


**Table 1: Specimen groups and information on their respective localities. Temperature and salinity ranges at Tès and Baie des Veys are from Lartaud et al. (2010b), and at Leucate from Andrisoa (2019).**





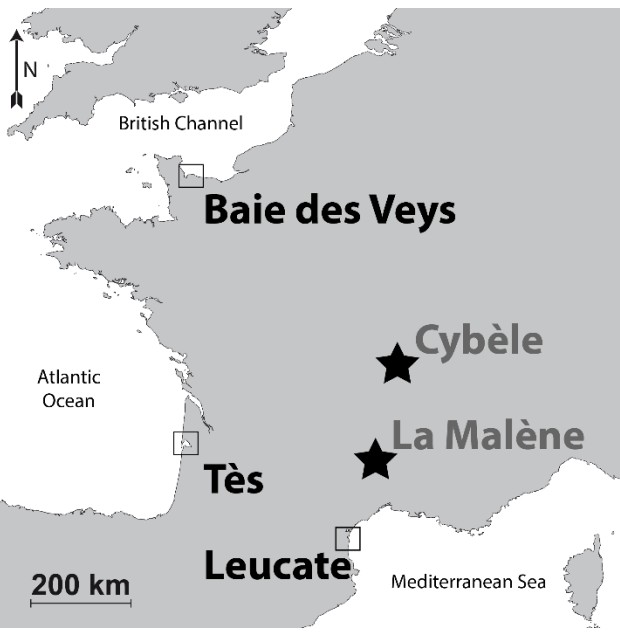

**Figure 1: Map of the localities of modern (squares) and archaeological (stars) specimens. Coordinates of the modern sites are indicated in Table 1.**







**Figure 2: Typical archaeological *Ostrea edulis* (a) and modern *Crassostrea gigas* (b) specimens. The umbo region (c) is cut following the dashed white line. Laser ablation craters (200 μm in diameter) are indicated by circles (on c). Multiple measurements have been performed on each shell, in both winter and summer parts. Red and blue circles represent measurements corresponding to summer and winter periods, respectively, based on cathodoluminescence.**

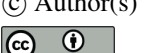



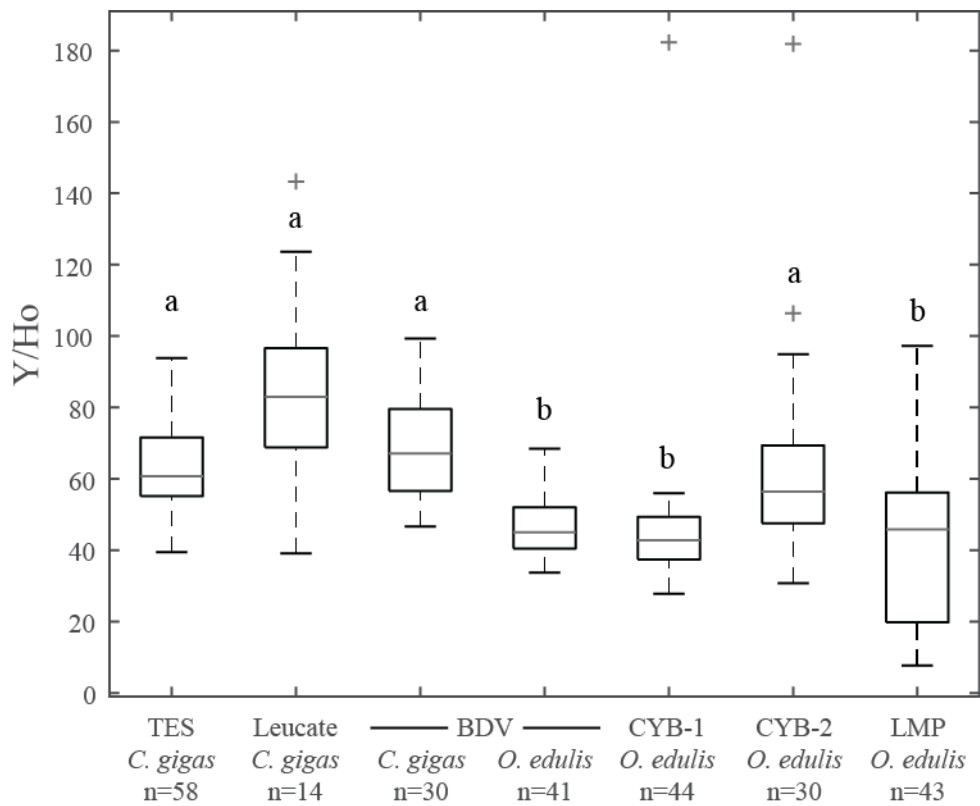






Figure 3: Boxplots of Y/Ho ratios for all groups. The letters on top of the boxes (a and b) identify the significant differences between groups from Kruskal-Wallis tests. Note that for the same locality at Baie des Veys, the Y/Ho ratios are significantly different depending on the species considered (BDV *C. gigas* and *O. edulis* groups). Grey bars represent median values, the lower and higher large black bars represent the 25th and 75th percentiles, respectively, and the lower and higher small black bars represent the minimum and maximum values not considered as outliers, respectively. Outliers are represented by grey crosses.

**Figure 4: Gradual decrease of REE abundances in oyster shells according to the REE atomic number, presented against Y. Values**
**are expressed in cubic root of abundances (in µg.g⁻¹) to approach normality. Measurements from *C. gigas* and *O. edulis* are**
**indicated with crosses and filled circles, respectively.**


EGU Open Access

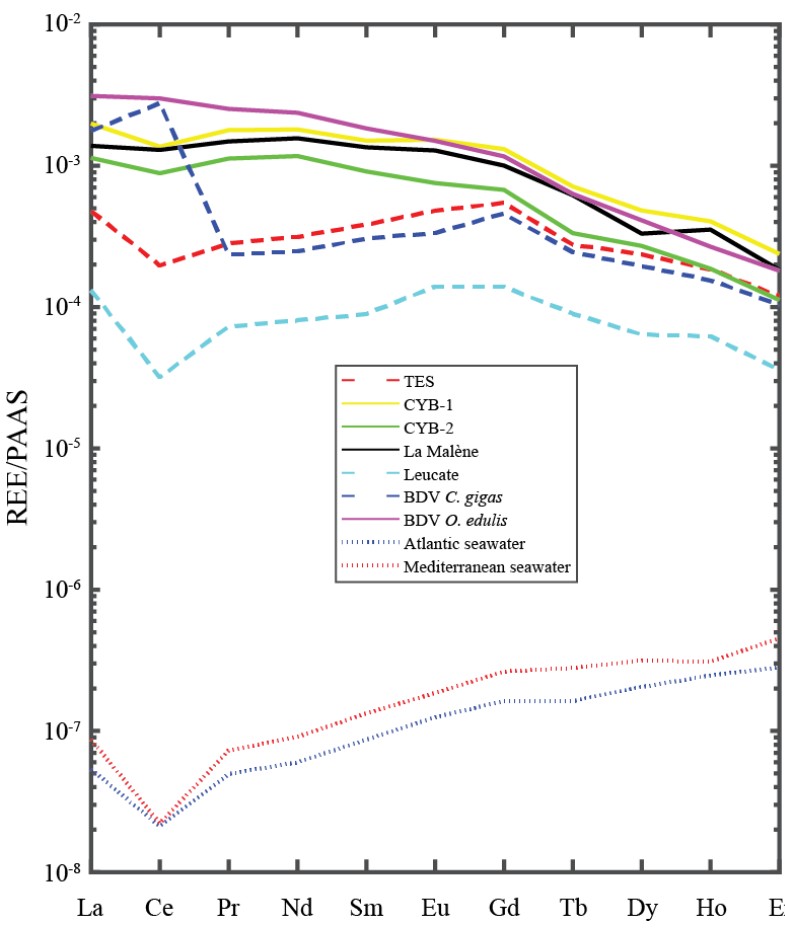

**Figure 5: REE median profiles for all groups of oyster specimens. *Crassostrea gigas* groups are symbolized with dashed lines and continuous lines for *O. edulis* groups. Seawater profiles of the Atlantic Ocean (van der Flierdt et al., 2012) and the Mediterranean Sea (Censi et al., 2004) are indicated for comparison. Values are normalized to Post-Archean Australian Shales (PAAS) according to McLennan (1989).**






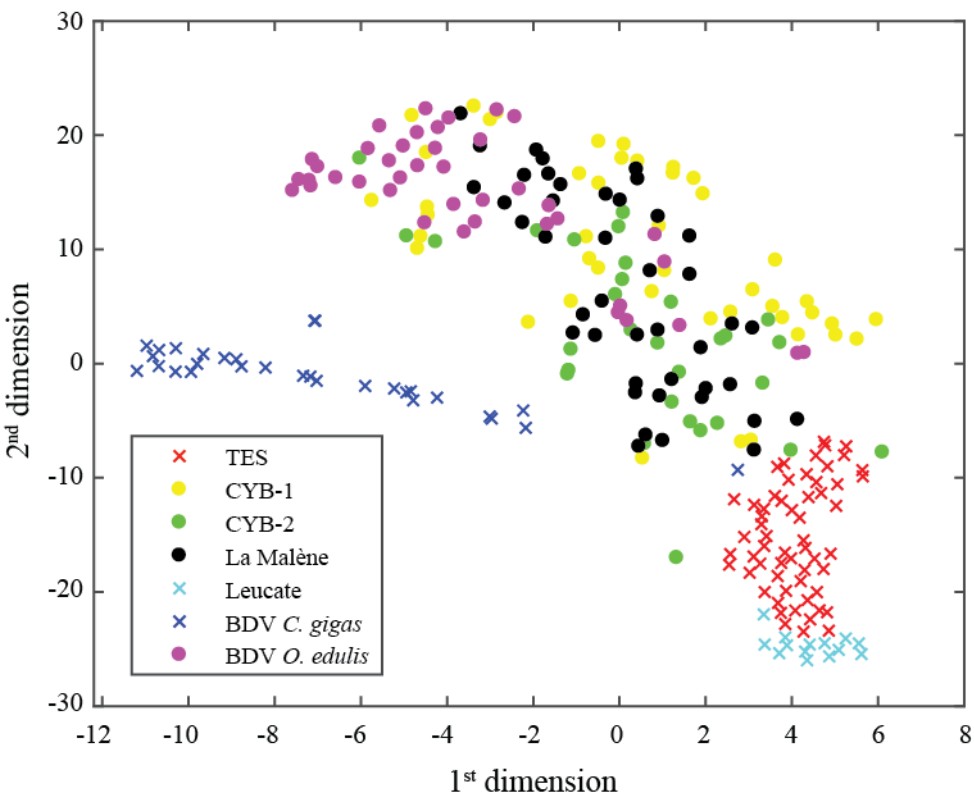

**Figure 6: Visualization of shell group partitioning using t-SNE applied to all REE and Y measurements as variables. Crosses and large dots refer to *C. gigas* and *O. edulis*, respectively.**