# Peer review of "Rare Earth Elements in oyster shells: provenance discrimination and potential vital effects"

_Biogeosciences, 2019_

## Referee Comment (RC1) · Anonymous Referee #1 · 31 Dec 2019

General comments

The authors set out to test whether the REE and Y compositions of bivalve shells can be used as provenance and environmental tracers. They used Laser Ablation-ICP-MS to measure modern oysters from three coastal locations and ancient oysters from two archaeological sites. The authors conclude that the Y/Ho ratio in C. gigas specimens gives information on their provenance but that this is not the case for O. edulis.

This study presents interesting data but has several major flaws.

1) Modern O. edulis specimens were only measured from one location, therefore there is no information on any differences in REY compositions in different locations for this species.

[Figure]

2) Only O. edulis specimens were measured in the archeological sites and, as stated above, these could only be compared with modern O. edulis from a single location.

3) There is no information about the measured or expected REY in the seawater at the different coastal sites and therefore it is unclear if measurable differences should be expected.

4) The authors attribute the similarities in REY compositions between the ancient O. edulis specimens to vital effects. This conclusion cannot be validated without addressing points 1 and 3 above.

I do not recommend publication unless these points are adequately addressed.

I have no further comments at this time and will wait until a revised version is submitted before adding specific comments and technical corrections.

---

## Referee Comment (RC2) · Anonymous Referee #2 · 5 Jan 2020

The authors of 'Rare Earth Elements in oyster shells: provenance discrimination and potential vital effects' aim to show how Rare Earth Elements (REE) and Yttrium (Y) conc. in bivalve shells could be used as a provencing and environmental monitoring tool. The authors use a combination of chemical analyses (LA-ICP-MS) and machine learning techniques (t-SNE) to discriminate bivalve specimens between sampled locations/species. Fingerprinting tools such as this are of increasing importance from a food safety standpoint, but the authors highlight their potential within an archaeological and monitoring context too. Despite the novelty of the method used herein, there are some points that I believe require attention prior to publication.

1) As mentioned by the first anonymous reviewer, a significant conclusion made by the authors of this manuscript was that O. edulis could not be discriminated spatially because of a lack of intraspecific variation in REE and Y concentrations. Yet, the authors did not test this between spatially disparate locations of the same age. I therefore do not believe this conclusion can be supported at this point.

2) In contrast to reviewer #1 I do not believe it necessary to gather water samples to elucidate whether a difference in REE and Y will be likely. Water samples will only provide a snapshot of site-specific conc. at any given time/place and inferences could not be made of the REE and Y concentrations within the slow forming carbonate shells. However, I would agree with reviewer #1 that the conclusions made surrounding O. edulis and vital effects here are not supported due to the limited number of sites assessed of the same time.

3) I believe from the outset, the aims of this study are not immediately clear. More emphasis could be placed in the introduction on the rational for this study, where it fits within seafood traceability or in regulatory capacity. Again, the implications of the authors' findings are also not well established in the discussion. The authors place a significant focus on the results at hand, which is of course important, but some emphasis should be placed on where this research now fits within its field (i.e. uses, pros, pitfalls and directions from here).

4) Overall, I found the presentation of results somewhat challenging to follow. For example, I found myself having to refer back to Table 1 often to remind myself of the 'Groups' - site, species and age of the specimens. Simple adjustments could streamline this, making it more user friendly for the reader (i.e. no. for site, code for time, code for species).

5) More information should be provided for quality assurance. Lines 163-167 provide details of this. I would encourage the authors to retain this section but also include a quality assurance table of obtained vs. expected for certified reference material (CRMs) used for simple reference.

6) The authors introduce too many abbreviations. REE is fine, but REY for Rare Earth

Elements and yttrium is superfluous. Consider REE and Y instead. Same for HREE and LREE, Heavy REE reads fine.

At this time, I would not recommend this manuscript for publication until these points are addressed.

---

## Author Comment (AC1) · 3 Feb 2020

We wish to thank the referee for accepting reviewing our manuscript. Our responses to the comments are listed below.

Referee's comment: 1. Modern O. edulis specimens were only measured from one location, therefore there is no information on any differences in REY compositions in different locations for this species.

Authors' reply: Initially, our manuscript used only one modern site with both species, and several archaeological groups of O. edulis shells. We aimed at highlighting the similarities between O. edulis shells from several modern and ancient localities, and, separately, between C. gigas shells from several (modern) localities, as well as the

differences between both species. As requested, we added in a new modern locality on the Atlantic Ocean coastline with both species. The results and interpretations are the same than those of our initial manuscript.

Referee's comment: 2. Only O. edulis specimens were measured in the archeological sites and, as stated above, these could only be compared with modern O. edulis from a single location.

Author's reply: Crassostrea gigas first appeared on the French coastlines during the 20th century. We are sorry we omitted this rather important fact in our initial manuscript. The addition of a second modern locality with O. edulis specimens confirms our previous interpretations: O. edulis specimens from all modern and ancient localities present the same composition in REE, while C. gigas shells have a distribution (using t-SNE) gathering in clusters depending on their origin.

Referee's comment: 3. There is no information about the measured or expected REY in the seawater at the different coastal sites and therefore it is unclear if measurable differences should be expected.

Authors' reply: We have no seawater available for the localities, but we do not see the benefit for our study. We wish to present differences in REE shell incorporation processes between oyster species that have been reared on the exact same locality and conditions (and hence, same seawater composition). We added a supplementary picture (Appendix A) to show the direct proximity of both species bred on sites. In the Methods section, we present the type of geological substrate for each watershed as an indication of sources of REE in addition to oceanic seawater. For example, the Leucate locality has only low-REE content carbonates, while the watershed of Baie des Veys contains magmatic rocks. We discuss the Y/Ho ratio from the literature, which is based on the assumption of different local REE content. Also, even without known seawater composition, our results indicate a strong locality influence for C. gigas specimens.

Referee's comment: 4. The authors attribute the similarities in REY compositions between the ancient O.edulis specimens to vital effects. This conclusion cannot be validated without addressing points 1 and 3 above.

Authors' reply: As requested, we now present measurements from a second modern locality. We agree with Referee #2 that seawater composition measurements are unnecessary and probably inadequate. We hope that the changes made on this revised manuscript and the added explanations are satisfactory.

---

## Author Comment (AC2) · 3 Feb 2020

First of all, we wish to thank the Referee for accepting to review our manuscript. Our replies are listed below.

Referee's comment: 1. As mentioned by the first anonymous reviewer, a significant conclusion made by the authors of this manuscript was that O. edulis could not be discriminated spatially because of a lack of intraspecific variation in REE and Y concentrations. Yet, the authors did not test this between spatially disparate locations of the same age. I therefore do not believe this conclusion can be supported at this point.

Authors' reply: We have added to the dataset a new site, Marennes-Oléron, with modern specimens from both species. Our new results are in accordance with our previous

interpretations.

Referee's comment: 2. In contrast to reviewer #1 I do not believe it necessary to gather water samples to elucidate whether a difference in REE and Y will be likely. Water samples will only provide a snapshot of site-specific conc. at any given time/place and inferences could not be made of the REE and Y concentrations within the slow forming carbonate shells. However, I would agree with reviewer #1 that the conclusions made surrounding O. edulis and vital effects here are not supported due to the limited number of sites assessed of the same time.

Authors' reply: Initially, our manuscript used only one modern site with both species, and several archaeological groups of O. edulis shells. We aimed at highlighting the similarities between O. edulis shells from several modern and ancient localities, and, separately, between C. gigas shells from several (modern) localities, as well as the differences between both species. As requested, we added in this revised version a new modern locality on the Atlantic Ocean coastline with both species. The results and interpretations are the same than those of our initial manuscript.

Referee's comment: 3. I believe from the outset, the aims of this study are not immediately clear. More emphasis could be placed in the introduction on the rational for this study, where it fits within seafood traceability or in regulatory capacity. Again, the implications of the authors' findings are also not well established in the discussion. The authors place a significant focus on the results at hand, which is of course important, but some emphasis should be placed on where this research now fits within its field (i.e. uses, pros, pitfalls and directions from here).

Authors' reply: Changes have been made to clarify our goals and conclusions. In particular, we now put more emphasis on seafood traceability, which is relevant to both modern and archaeological contexts.

Referee's comment: 4. Overall, I found the presentation of results somewhat challenging to follow. For example, I found myself having to refer back to Table 1 often to remind

myself of the 'Groups' - site, species and age of the specimens. Simple adjustments could streamline this, making it more user friendly for the reader (i.e. no. for site, code for time, code for species).

Authors' reply: We have changed the names of the groups in the text and on the figures. As suggested by the reviewer, we now use a code indicating modern (Mod-) or ancient (Anc-) specimens as well as the locality and species names. Modern localities are now presented in the Method sections and in the figures from North to South.

Referee's comment: 5. More information should be provided for quality assurance. Lines 163-167 provide details of this. I would encourage the authors to retain this section but also include a quality assurance table of obtained vs. expected for certified reference material (CRMs) used for simple reference.

Authors' reply: We have added these details in Appendix B.

Referee's comment: 6. The authors introduce too many abbreviations. REE is fine, but REY for Rare Earth Elements and yttrium is superfluous. Consider REE and Y instead. Same for HREE and LREE, Heavy REE reads fine.

Authors' reply: We have removed all occurrences of REY, HREE and LREE from the manuscript.

---

## Author Response (AR1)

Reviewer #1 comments

The authors set out to test whether the REE and Y compositions of bivalve shells can be used as provenance and environmental tracers. They used Laser Ablation-ICP-MS to measure modern oysters from three coastal locations and ancient oysters from two archaeological sites. The authors conclude that the Y/Ho ratio in *C. gigas* specimens gives information on their provenance but that this is not the case for *O. edulis*.

This study presents interesting data but has several major flaws.

1) Modern *O. edulis* specimens were only measured from one location, therefore there is no information on any differences in REY compositions in different locations for this species.

*Initially, our manuscript used only one modern site with both species, and several archaeological groups of* O. edulis *shells. We aimed at highlighting the similarities between* O. edulis *shells from several modern and ancient localities, and, separately, between* C. gigas *shells from several (modern) localities, as well as the differences between both species. As requested, we added in a new modern locality on the Atlantic Ocean coastline (Marennes-Oléron) with both species. The results and interpretations are the same than those of our initial manuscript.*

2) Only *O. edulis* specimens were measured in the archeological sites and, as stated above, these could only be compared with modern *O. edulis* from a single location.

Crassostrea gigas *first appeared on the French coastlines during the 20th century. We are sorry we omitted this rather important fact in our initial manuscript. We now indicate this L. 72-73. The addition of a second modern locality with* O. edulis *specimens confirms our previous interpretations:* O. edulis *specimens from all modern and ancient localities present the same composition in REE, while* C. gigas *shells have a distribution (using t-SNE) gathering in clusters depending on their origin.*

3) There is no information about the measured or expected REY in the seawater at the different coastal sites and therefore it is unclear if measurable differences should be expected.

*We have no seawater available for the localities, but we do not see the benefit for our study. We wish to present differences in REE shell incorporation processes between oyster species that have been reared on the exact same locality and conditions (and hence, same seawater composition). We added a supplementary picture (Appendix A) to show the direct proximity of both species bred on sites. In the Methods section, we present the type of geological substrate for each watershed as an indication of sources of REE in addition to oceanic seawater. For example, the Leucate locality has only low-REE content carbonates, while the watershed of Baie des Veys contains magmatic rocks. We discuss the Y/Ho ratio from the literature, which is based on the assumption of different local REE content. Also, even without known seawater composition, our results indicate a strong locality influence for* C. gigas *specimens.*

4) The authors attribute the similarities in REY compositions between the ancient *O.edulis* specimens to vital effects. This conclusion cannot be validated without addressing points 1 and 3 above.

*As requested, we now present measurements from a second modern locality. We agree with Reviewer #2 that seawater composition measurements are unnecessary and probably inadequate. We hope that the changes made on this revised manuscript and the added explanations are satisfactory.*

Reviewer #2 comments

The authors of 'Rare Earth Elements in oyster shells: provenance discrimination and potential vital effects' aim to show how Rare Earth Elements (REE) and Yttrium (Y) conc. in bivalve shells could be used as a provencing and environmental monitoring tool. The authors use a combination of chemical analyses (LA-ICP-MS) and machine learning techniques (t-SNE) to discriminate bivalve specimens between sampled locations/species. Fingerprinting tools such as this are of increasing importance from a food safety standpoint, but the authors highlight their potential within an archaeological and monitoring context too. Despite the novelty of the method used herein, there are some points that I believe require attention prior to publication.

1) As mentioned by the first anonymous reviewer, a significant conclusion made by the authors of this manuscript was that *O. edulis* could not be discriminated spatially because of a lack of intraspecific variation in REE and Y concentrations. Yet, the authors did not test this between spatially disparate locations of the same age. I therefore do not believe this conclusion can be supported at this point.

*We have added to the dataset a new site, Marennes-Oléron, with specimens from both species. Our new results are in accordance with our previous interpretations.*

2) In contrast to reviewer #1 I do not believe it necessary to gather water samples to elucidate whether a difference in REE and Y will be likely. Water samples will only provide a snapshot of site-specific conc. at any given time/place and inferences could not be made of the REE and Y concentrations within the slow forming carbonate shells. However, I would agree with reviewer #1 that the conclusions made surrounding *O. edulis* and vital effects here are not supported due to the limited number of sites assessed of the same time.

*Initially, our manuscript used only one modern site with both species, and several archaeological groups of* O. edulis *shells. We aimed at highlighting the similarities between* C. gigas *shells from several (modern) localities, and, separately, between* O. edulis *shells from several modern and ancient localities, as well as the differences between both species. As requested, we added in this revised version a new modern locality on the Atlantic Ocean coastline with both species. The results and interpretations are the same than those of our initial manuscript.*

3) I believe from the outset, the aims of this study are not immediately clear. More emphasis could be placed in the introduction on the rational for this study, where it fits within seafood traceability or in regulatory capacity. Again, the implications of the authors' findings are also not well established in the discussion. The authors place a significant focus on the results at hand, which is of course important, but some emphasis should be placed on where this research now fits within its field (i.e. uses, pros, pitfalls and directions from here).

*Changes have been made to clarify our goals and conclusions. In particular, we now put more emphasis on seafood traceability, which is relevant to both modern and archaeological contexts.*

4) Overall, I found the presentation of results somewhat challenging to follow. For example, I found myself having to refer back to Table 1 often to remind myself of the 'Groups' - site, species and age of the specimens. Simple adjustments could streamline this, making it more user friendly for the reader (i.e. no. for site, code for time, code for species).

*We have changed the names of the groups in the text and on the figures. As suggested by the reviewer, we now use a code indicating modern (Mod-) or ancient (Anc-) specimens as well as the locality and species names. Modern localities are now presented in the Method sections and in the figures from North to South.*

5) More information should be provided for quality assurance. Lines 163-167 provide details of this. I would encourage the authors to retain this section but also include a quality assurance

table of obtained vs. expected for certified reference material (CRMs) used for simple reference.

*We have added these details in Appendix B.*

6) The authors introduce too many abbreviations. REE is fine, but REY for Rare Earth Elements and yttrium is superfluous. Consider REE and Y instead. Same for HREE and LREE, Heavy REE reads fine.

*We have removed all occurrences of REY, HREE and LREE from the manuscript.*

[revised manuscript text omitted]

Déplacé (insertion) [2]
Déplacé (insertion) [3]
Déplacé (insertion) [4]
Déplacé (insertion) [5]
Déplacé (insertion) [6]
Déplacé (insertion) [7]
Déplacé (insertion) [8]
Déplacé (insertion) [9]
Déplacé (insertion) [10]
Déplacé (insertion) [11]
Déplacé (insertion) [12]
Déplacé (insertion) [13]
Déplacé (insertion) [14]
Déplacé (insertion) [15]
Déplacé vers le haut [13]: Leucate, Aude (Mediterranean

[Figure]

730

**Figure 1: Map of the localities of modern (squares) and archaeological (stars) specimens. Coordinates of the modern sites are indicated in Table 1.**

[Figure]

**Figure 2: Typical archaeological *Ostrea edulis* (a) and modern *Crassostrea gigas* (b) specimens. The umbo region (c) is cut following the dashed white line. Laser ablation craters (200 μm in diameter) are indicated by circles (on c). Multiple measurements have been performed on each shell, in both winter and summer parts. Red and blue circles represent measurements corresponding to summer and winter periods, respectively, based on cathodoluminescence.**

[Figure]

**Figure 3: Boxplots of Y/Ho ratios for all groups. The letters on top of the boxes (a and b) identify the significant differences between groups from Kruskal-Wallis tests. Note that for the same locality at Baie des Veys (BDV) and Marennes-Oléron (MO), the Y/Ho ratios are significantly different depending on the species considered (*C. gigas* and *O. edulis* groups). Grey bars represent median values, the lower and higher large black bars represent the 25th and 75th percentiles, respectively, and the lower and higher small black bars represent the minimum and maximum values not considered as outliers, respectively. Outliers are represented by grey crosses.**

[Figure]

**Figure 4: Gradual decrease of REE abundances in oyster shells according to the REE atomic number, presented against Y. Values are expressed in cubic root of abundances (in µg.g⁻¹) to approach normality. Measurements from *C. gigas* and *O. edulis* are indicated with crosses and filled circles, respectively.**

750

[Figure]

[Figure]

**Figure 5: REE median profiles for all groups of oyster specimens.** *Crassostrea gigas* **groups are symbolized with dashed lines and continuous lines for** *O. edulis* **groups. Seawater profiles of the Atlantic Ocean (van der Flierdt et al., 2012) and the Mediterranean Sea (Censi et al., 2004) are indicated for comparison. Values are normalized to Post-Archean Australian Shales (PAAS) according to McLennan (1989).**

755

[Figure]

[Figure]

**Figure 6: Visualization of shell group partitioning using t-SNE applied to all REE and Y measurements as variables. Crosses and large dots refer to *C. gigas* and *O. edulis*, respectively.**

---

## Author Response (AR2)

Rebuttal (2) for Mouchi et al.: 'Rare Earth Elements in oyster shells: provenance discrimination and potential vital effects'

**General comments.**

The authors have improved their manuscript and substantiated their conclusions with the inclusion of modern specimens of *O. edulis* from an additional site.

However, I think that the introduction and discussion are missing some important information on REEs in seawater and biogenic carbonate material that would help to assess whether the chosen method is suitable or not. The authors should consider the following:

- Zaky et al (2015) highlighted that strict cleaning procedures were required in order to reproduce seawater-like REE patterns in brachiopod analyses. They checked this by reporting the distribution coeffecients. I accept that Mouchi and co-authors want to use a rapid sample processing approach in their study but this should be assessed more thoroughly. If the kDs are widely different from other mollusc studies, then it may be the method rather than the specimens themselves that are causing the similarities between disparate groups.

Response: We thank the reviewer for his/her time on our manuscript. We however disagree regarding the concerns about the potential contamination of our specimens for two reasons:

- Zaky et al. (2015) tested different procedures for solution-based ICP-MS measurements and indeed highlighted the benefit of both extensive physical and chemical cleaning prior dissolution of the sample and analysis due to surface contaminants. The physical cleaning they suggest precisely aims at exposing internal (preserved and uncontaminated) shell areas. We however perform our analyses by LA-ICP-MS on internal, preserved sections of the shells. We precisely indicated this in our initial manuscript L. 151-154: "thick sections (approx. 750 µm thick) were manufactured to expose the preserved internal structures (Figure 2), in order to perform geochemical analyses on this protected region, away from shell surface organic or chemical contaminants. An extensive chemical cleaning of the section surfaces, as advised by Zaky et al. (2015), was not performed as the analytical surface was preserved from any external contaminants over the history of the shell."
- Also, if it were indeed contaminants that are causing the similarities between *O. edulis* groups of various geographic origin, we would have the same observation for *C. gigas* groups, especially those collected on the same sites than *O. edulis* specimens. On the contrary, although all specimens have been prepared in an identical fashion, the *C. gigas* specimens show a region-specific REE composition, and the *O. edulis* have systematically the same REE composition whatever the region of origin, even for specimens of both species from the same sites. If the method was inadequate, it would be for all observations, which is not the case. We are therefore strongly confident about our method.

Minor comments

**Some specific comments are listed below.**

Line 37. Unfortunately, I don't have access to the Jouanneau et al paper but from the abstract, they measured REE in sediments, rather than seawater. From the sentence beginning in line 35, it sounds like the authors wish to compare this with studies that reconstruct REE compositions in seawater. Did Jouanneau et al. measure the sediment transported by the river, or did they measure a seawater signal? This needs to be made clear because the REE composition of river water undergoes major changes upon reaching the estuary (see work of Sholkovitz especially), so it would be incorrect to say that the dissolved fraction on the continental shelf reflects the composition of contributor rivers. **Our sentence was indeed misleading. We removed this sentence.**

Line 180. Do you have a reference for the BCR-2 reference material where the composition is listed?

The reference of Jochum et al. (2005) was already indicated in Appendix B, in which we indicate our measurements and the expected values. We have nevertheless added this reference in the manuscript.

Lines 223-224. The start of the sentence is overly complicated. Consider simplifying this to "The Y/Ho ratios in the present study (Figure 3) do not....".

Lines 228, 233, 244 and 235. Look again at the connectors used at the beginning of the sentences in order to highlight whether something is the same as or different from the observation in the previous sentence. It is hard to decipher what is reported here, especially with the very long bracketed sections.

We did our best to find better connectors. We hope it is clearer now.

Lines 307-309. I agree with this sentence and would point out that as a result, it is not accurate to say that 'strong vital effects' are responsible for the Y/Ho ratios in the oysters (line 285). It is not known whether this is a 'vital effect' or just "passive" incorporation into the shell.

"Passive" incorporation could be at play for only one of the two species. We consider as 'vital effects' this difference of chemical incorporation in the shell between species. We have however agreed to follow the reviewer's suggestion to be less specific in the conclusion.

**All other corrections required from the reviewer have been made:**

Lines 18 – 20. This sentence should be divided into two in order to make more sense. E.g. "Here, we present a dataset of 297 measurements of REE and Y abundances by LA-ICP-MA from two oyster species (*Crassostrea gigas* and *Ostrea edulis*). We measured a total of 49 oyster specimens from six locations in France (Atlantic Ocean and Mediterranean Sea)." Line 26. Remove "and are not adapted for *O. edulis* as this is a repeat of the information in the previous sentence.

Line 29. Replace "form a group gathering" with "are a group of". Also, add "The" to the start of the sentence.

Line 30. There are important exceptions to the statement that the REE have similar properties and chemical behavior – what about Ce and Eu?

Line 30. Remove 'the' before atmospheric fallout.

Line 33. The end of the sentence appears to be missing. Complexation with what?

Line 38. Remove "from REE", as Y isn't a REE and this sentence makes sense without saying this.

Lines 45-47. It is incorrect to say that the "seawater" composition of REE and Y is recorded in carbonate materials, as it is always altered during uptake/adsorption (e.g. Osborne et al., 2017).

Lines 55-57. This relates to the previous comment – the REE composition recorded in carbonates (not just corals) is altered during uptake/adsorption.

Line 64. Intraspecific variations and potential seasonal fluctuations in what exactly?

Line 72. Is "British Channel" the international name for this stretch of water? I know it as the "English Channel".

Line 81. "in" the shells.

Line 85. "is" the most commonly found.

Line 87. Move "respectively" to the end of the sentence.

Lines 89-91. No need to capitalize western, northern, north etc.

Line 90. "Its surface area is approximately...". Also, remove "The" at the beginning of the next sentence.

Line 96. Amongst "these". Either "a" rearing experiment, or "the rearing experiment of Lartaud et al.....".

Line 98. "their stay at this site"

Line 99. Move "respectively" to the end of the sentence.

Line 106. "consists of", "to a lesser extent".

Line 113. Remove "of" before 14.

Line 114. "the hydrology is a balance between"

Lines 117-118. It is unclear what "likely extended to 60 km far from the pool" means.

Line 129. Please add some references to this statement.

Line 135. "is comprised of" rather than "is constituted by".

Line 155. "as in any"

Line 163. "Only the Leucate". First occurrence of the abbreviation "CL". This needs to be defined in line 162.

Line 176. It is unclear what "different year periods" means. Do you mean seasons?

Line 179. "prior to".

Line 184. "made" rather than "executed".

Line 195. What is the "it" at the start of the sentence? The cophenetic correlation?

Line 198. Replace "performed from" with "of".

Line 202. "The idea of the t-SNE method is to..."

Line 223. It doesn't make sense to say that something is commonly used as a "potential" provenance proxy.

Line 242. You need to define what you mean by "dispersion" i.e. what you plot in figure 4 (abundance relative to Y).

Line 245. It is more usual to say "middle" REE.

Line 247. "the Mod\_LEU\_Cgig group"

Line 251. Replace "particularity" with "feature" or "pattern".

Lines 265 and 267. This sentence is overly complicated. Simplify by removing "it appears that", "on the one hand" and "on the other hand".

Lines 276 and 277. "and other REE" is vague – can you be more specific? You want to emphasize that the heavy REE have lower bioavailability. (Gd is a middle REE).

Line 291. "in providing" rather than "to provide".

Lines 314-322. This seems an overly long explanation. It would be simpler to say that the Gd anomaly is only seen in the modern C. gigas specimens at two locations because they are the only coastal regions with a major city in the watershed, and then that it is not seen in the *O*. *edulis* specimens from these locations.

Lines 322-324. This statement needs some references.

Line 324. "discriminate between".

Line 243. The last part of the sentence doesn't make sense.

Lines 345-346. It isn't possible to generalize to "other oyster species" as only one other was tested. It is more accurate to say that "intraspecific vital effects prevent its application in *O. edulis*, whose specimens..."

Lines 348-349. The last two sentences can be omitted.

Line 351. Again, it is not possible to say whether these are 'vital effects' or not. It would be more accurate to say that there are many factors that affect the incorporation of REEs into oyster shells.

Line 354. "in identifying"

Line 355. "is ineffective"

Line 357. "Reconstruction of the provenance"

**Rare Earth Elements in oyster shells: provenance discrimination and potential vital effects**

Vincent Mouchi1, Camille Godbillot1, Vianney Forest2, Alexey Ulianov3, Franck Lartaud4, Marc de Rafélis5, Laurent Emmanuel1, Eric P. Verrecchia6

[revised manuscript text omitted]